# AffordBot: 3D Fine-grained Embodied Reasoning via Multimodal Large Language Models

**Xinyi Wang[1,*], Xun Yang[1,†], Yanlong Xu[1], Yuchen Wu[2], Zhen Li[3], Na Zhao[2,†]**

[1] University of Science and Technology of China   [2] Singapore University of Technology and Design
[3] Chinese University of Hong Kong, Shenzhen

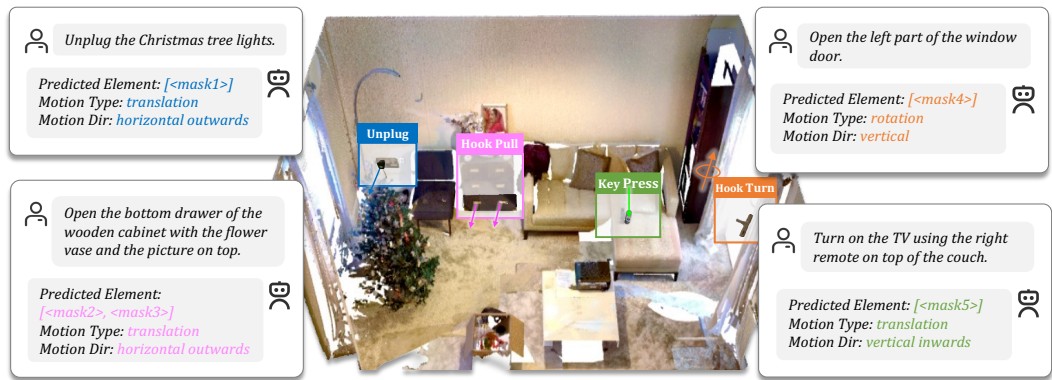

Figure 1: We propose fine-grained 3D embodied reasoning: given a 3D scene and a language task instruction, the agent must identify relevant affordance elements and predict a structured triplet for each: its 3D mask, motion type, and motion axis direction.

## Abstract

Effective human-agent collaboration in physical environments requires understanding not only *what* to act upon, but also *where* the actionable elements are and *how* to interact with them. Existing approaches often operate at the object level or disjointedly handle fine-grained affordance reasoning, lacking coherent, instruction-driven grounding and reasoning. In this work, we introduce a new task: Fine-grained 3D Embodied Reasoning, which requires an agent to predict, for each referenced affordance element in a 3D scene, a structured triplet comprising its spatial location, motion type, and motion axis, based on a task instruction. To solve this task, we propose AffordBot, a novel framework that integrates Multimodal Large Language Models (MLLMs) with a tailored chain-of-thought (CoT) reasoning paradigm. To bridge the gap between 3D input and 2D-compatible MLLMs, we render surround-view images of the scene and project 3D element candidates into these views, forming a rich visual representation aligned with the scene geometry. Our CoT pipeline begins with an active perception stage, prompting the MLLM to select the most informative viewpoint based on the instruction, before proceeding with step-by-step reasoning to localize affordance elements and infer plausible interaction motions. Evaluated on the SceneFun3D dataset, AffordBot achieves state-of-the-art performance, demonstrating strong generalization and physically grounded reasoning with only 3D point cloud input and MLLMs. Our code is available at `https://github.com/hannahwxy/AffordBot`.

---

[*]This work was carried out during Xinyi's visit to the IMPL Lab at SUTD.
[†]Corresponding authors.

# 1 Introduction

For intelligent agents to collaborate effectively with humans and operate autonomously in complex 3D physical worlds, they must perceive and interact with their surroundings at a fine-grained, actionable level [1, 2, 3]. This requirement aligns with the concept of "affordance", originally introduced in ecological psychology [4], which describes how elements in the environment offer possibilities for action. For example, to execute an instruction like "*unplug the Christmas tree lights*", an agent must recognize and attend to the fine details of the plug and reason about the interaction they afford, rather than merely identifying the larger object, such as the lights. Meeting this challenge requires agents to understand not only *what* elements in a scene afford interaction, but also *where* they are located and *how* to manipulate them. Such fine-grained embodied understanding is essential for grounded task execution in real-world environments [5, 6, 7, 8, 9, 10, 11, 12, 13].

While recent developments in multimodal large language models (MLLMs) [14, 15, 16, 17, 18, 19, 20, 21] and 3D scene understanding [22, 23, 24, 25, 26, 27] have advanced object-centric 3D perception, existing approaches [28, 29, 30, 31, 32, 33] stop at high-level object recognition and spatial grounding. However, they often overlook finer-grained structures required to infer how parts of objects afford specific interactions. SceneFun3D [34] makes a notable step forward by introducing benchmarks for fine-grained affordance grounding and motion estimation. However, it treats these subtasks in isolation and assumes instruction-agnostic motion prediction, requiring agents to infer motion parameters for all functional parts regardless of task context, limiting its applicability in instruction-conditioned scenarios.

To address these limitations, we propose a unified and instruction-conditioned task: **Fine-grained 3D Embodied Reasoning**, which jointly performs 3D affordance grounding and motion estimation based on a natural language instruction. Specifically, the task is formulated as a structured triplet prediction problem: for each referenced affordance element, the agent predicts a triplet comprising *affordance mask*, *motion type*, and *motion axis direction*. This formulation explicitly couples spatial grounding and interaction reasoning under natural language guidance, forming a coherent inference pipeline tailored for instruction-conditioned embodied tasks.

As a solution, we introduce **AffordBot**, a novel framework that integrates 3D geometric information with the reasoning capabilities of MLLMs. Unlike prior work [34, 35, 36, 37, 38] that relies on video-based inputs, which incur high computational overhead by processing redundant visual frames and often suffer from viewpoint limitations, AffordBot operates directly on 3D point clouds. However, MLLMs are inherently designed for 2D input and general-purpose reasoning, presenting a significant challenge when applying them to 3D spatial tasks that require physical grounding.

To bridge the modality gap between 3D input and 2D-native MLLMs, AffordBot begins by constructing a rich multimodal representation of the 3D scene. Specifically, we render surround-view images from the 3D point cloud and project structured 3D affordance candidates onto these images, establishing explicit 3D-to-2D correspondences. This enables dense and spatially aligned visual context to be provided to the MLLM, without relying on redundant video streams.

On top of this foundation, we develop a task-specific chain-of-thought (CoT) reasoning paradigm that systematically guides the MLLM through physically grounded, step-by-step logical inference. The process begins with an active perception phase, which we specifically designed to empower the MLLM to effectively interpret the task instruction and autonomously select the most informative viewpoint. This initial "*observe*" step improves reasoning focus by reducing input redundancy and emphasizing task-relevant visual cues. Subsequently, the selected viewpoint anchors the model's reasoning process. Then the MLLM is guided through two distinct reasoning stages: *affordance grounding*, where it localizes the target part in the scene, and *interaction inference*, where it predicts the motion type and axis direction based on the scene context and instruction. By conditioning every step on spatial input and task intent, our CoT paradigm enables physically plausible and semantically aligned reasoning, enhancing the agent's embodied intelligence.

We make three key contributions: (1) We introduce a new task formulation for fine-grained, task-driven embodied reasoning, 3D affordance grounding and motion estimation as structured triplet prediction from natural language. (2) We present AffordBot, a novel framework that integrates 3D perception and MLLM-based reasoning via holistic multimodal representation construction and tailored chain-of-thought process. (3) We achieve state-of-the-art results on SceneFun3D, validating the effectiveness of our approach in physically grounded, instruction-conditioned 3D reasoning.

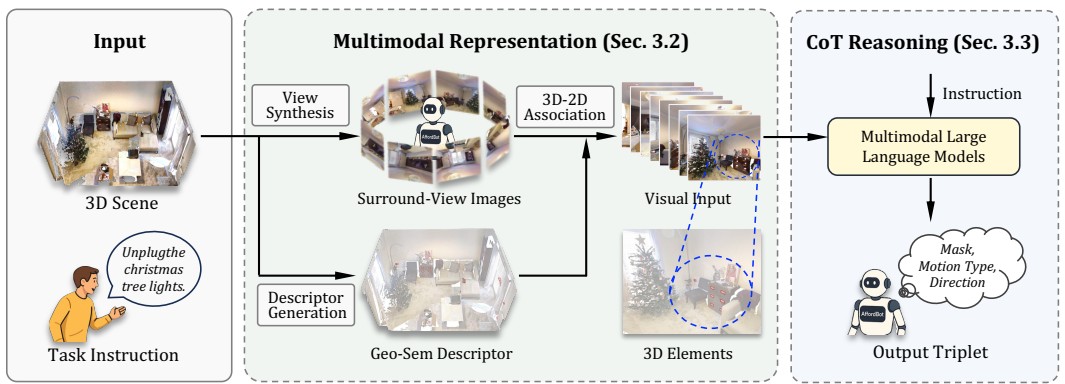

Figure 2: **AffordBot Overview**. Our method first constructs a holistic multimodal representation designed to bridge 3D scenes with 2D-native MLLMs. This process involves view synthesis, extraction of geometric-semantic descriptors, and their association. Then, our designed Chain-of-Thought (CoT) paradigm guides the MLLM to ultimately predict a structured triplet for the task.

## 2 Related Work

**Affordance Understanding.** Understanding affordances, the action possibilities offered by the environment [5, 39, 40, 41, 42, 43, 44, 45] , is crucial for robot interaction in 3D scenes. SceneFun3D [34] introduced a challenging task and dataset for grounding referred affordance elements based on task descriptions within complex 3D indoor environments, unlike earlier works focusing on simpler settings [5, 6, 7, 8, 9, 10, 12, 13]. Fun3DU [37] further leveraged VLMs [46] and a universal segmentation model [47] to parse instructions and localize the target elements in video frames.

**3D Motion Estimation.** 3D motion enables agents to predict and comprehend how objects move and can be manipulated [48, 49, 50, 51, 52, 53, 54, 55, 56, 57]. Earlier research efforts often focused on estimating the articulated motion and mobility of individual interactable objects with predefined structures, such as hinged parts [48, 49, 50]. These approaches typically rely on analyzing the geometric structure of those individual objects to infer their motion properties. In contrast, SceneFun3D [34] broadened this by formulating a scene-level motion estimation task across all affordance elements, providing a dataset for comprehensive evaluation.

**MLLMs for 3D Understanding.** Multimodal large language models (MLLMs) [14, 15, 16, 17, 18, 19] are being applied to 3D understanding through two main approaches: developing native 3D-aware models [30, 58, 59, 31, 60] for direct processing of spatial data, and adapting existing 2D VLMs [61, 62, 63, 64] by transforming 3D data into 2D representations. These efforts highlight MLLMs' potential for enhancing 3D visual understanding and semantic reasoning. Building upon these, we leverage the MLLM empowered by the tailored chain-of-thought paradigm for fine-grained 3D embodied reasoning, jointly tackling affordance grounding and motion estimation tasks.

## 3 Methodology

We introduce a new task termed **Fine-grained 3D Embodied Reasoning**, which aims to equip embodied agents with the ability to interpret natural linguistic instructions and reason about actionable elements in complex 3D environments. Given a 3D scene $\mathcal{S}$ represented as a point cloud and a natural language task instruction $\mathcal{T}$ describing a human-intended interaction (*e.g.*, "open the left part of the window door"), the agent is required to predict a set of **structured triplets** $\{(\mathbf{M}_i, \mathbf{t}_i, \mathbf{a}_i)\}_{i=1}^{N}$, where each triplet corresponds to a referenced affordance element in the scene. Note that N=1 when only one unique element is referenced in the instruction. $\mathbf{M}_i$ indicates a 3D instance mask identifying the spatial region of the element involved in the interaction; $\mathbf{t}_i \in \mathcal{T}$ denotes the motion type (*e.g.*, "Translation"); $\mathbf{a}_i \in \mathcal{A}$ denotes the motion axis (*e.g.*, "Horizontal outwards") representing the axis along which the motion occurs. The task requires *joint perception and reasoning* over geometry, semantics, and language intent, and presents challenges in grounding ambiguous task references, understanding object affordances, and predicting physically plausible interaction cues in a 3D space.

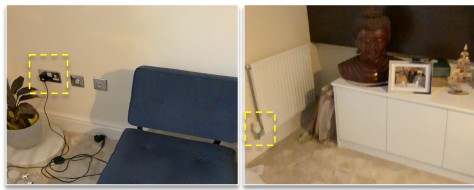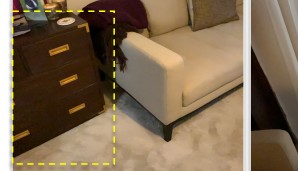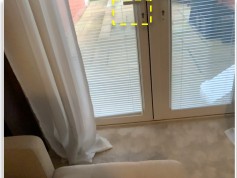

| (a) Lack of Context. | (b) Incomplete target coverage. |

Figure 3: **Illustrations of video-based method limitations:** (a) Instructions like *"Unplug the Christmas tree lights"* or *"Adjust the room's temperature using the radiator dial next to the curtain"* require anchors (e.g., Christmas tree, curtain) that are missing from the limited video frame. (b) Target objects or parts (*e.g.*, cabinet, door handle) in instructions like *"Open the bottom drawer of the wooden cabinet..."* or *"Open the left part of the window door"* are partially visible within the frame.

### 3.1 AffordBot Overview

To address the fine-grained 3D embodied reasoning task, we present AffordBot, a framework that leverages MLLMs to enable instruction-conditioned reasoning over 3D point cloud scenes. An overview of the framework is shown in Fig. 2. AffordBot integrates 3D perception with vision-language reasoning through two key components: 1) a holistic multimodal representation (Sec.3.2) that bridges the modality gap between 3D input and the 2D-native input format of MLLMs, and 2) a chain-of-thought reasoning paradigm (Sec. 3.3) that enables interpretable and accurate prediction.

For the first component, we begin by generating a set of surround-view images to effectively capture the 3D scene. Following this, we extract geometric-semantic descriptors from the 3D scene and project them onto the rendered views by an adaptive labeling strategy, establishing robust 3D-to-2D association. This representation effectively eliminates traditional video processing bottlenecks while preserving comprehensive information for downstream reasoning, as illustrated in green in Fig. 2.

Based on this constructed representation and the given instruction, the MLLM engages in a tailored chain-of-thought process to actively perceive and select the most informative view, localize the target element, and infer its required motion. By decomposing the task into a sequence of interpretable steps, our method enables the MLLM to perform robust and physically grounded inference for complex embodied tasks, depicted in blue in Fig. 2.

### 3.2 Holistic Multimodal Representation

In this section, we construct the holistic multimodal representation foundational for 2D MLLM reasoning. We design an enriched visual synthesis approach using dynamic surround-view generation to overcome limitations of traditional video data. Next, we describe the extraction and representation of 3D geometry and semantics via geometry-semantic descriptors. Finally, we establish the 3D-2D associations by projecting 3D information onto the generated 2D views with adaptive labeling.

**Enriched Visual Synthesis.** Bridging the gap between 3D input and 2D MLLMs is nontrivial, primarily because it requires establishing accurate and robust associations between 3D structures and their corresponding 2D visual representations.

Existing methods typically rely on video sequences collected from datasets [37, 34, 59]. However, these methods face fundamental limitations: due to the limited field of view, it is often difficult to simultaneously capture the target and its associated anchors within the same frame, as shown in Fig. 3. Furthermore, the process of extracting key information from a large number of frames is both time-consuming and bottlenecks the final accuracy.

To overcome the limitations of static video frames, we propose a dynamic surround-view generation strategy. Inspired by human visual exploration of unfamiliar environments, our robot performs a 360° horizontal panoramic scan centered on the scene's central viewpoint. This scan produces a set of $N$ candidate views $\mathcal{V} = \{V_1, \ldots, V_N\}$, where the $i$-th view $V_i$ is captured at a rotation angle $\theta_i = (i-1)\frac{2\pi}{N}$.

Compared to relying on traditional video data, this method provides a more comprehensive field of view, thereby capturing more scene information. This effectively alleviates the problem of

missing information caused by a limited field of view and incomplete coverage typical of traditional video data obtained through random sampling. Furthermore, by scanning each 3D scene to acquire a corresponding set of high-quality views, we eliminate the overhead of performing keyframe extraction or detection for each task instruction, thereby completely removing the time processing overhead and accuracy bottleneck associated with analyzing redundant video frames. This ability enables agents to efficiently acquire comprehensive, high-quality visual context.

**Geometry-Semantic Descriptors.** For the input 3D scene $\mathcal{P}$, our method employs instance segmentation [65] to extract affordance elements, and encodes their geometric and semantic features for downstream reasoning.

During training, we optimize segmentation quality by combining Dice loss to encourage region-level alignment and cross-entropy loss for accurate point-wise classification. The overall training objective is defined as:

$$\mathcal{L}_{\text{total}} = \lambda_1 \cdot \mathcal{L}_{\text{Dice}} + \lambda_2 \cdot \mathcal{L}_{\text{CE}}, \tag{1}$$

where $\mathcal{L}_{\text{Dice}}$ denotes the Dice loss and $\mathcal{L}_{\text{CE}}$ denotes the cross-entropy loss. The weights $\lambda_1$ and $\lambda_2$ balance the trade-off between the region-level and point-level supervision.

To handle the challenge of small element segmentation, we implement the coarse-to-fine curriculum strategy from [34] with progressive ground-truth mask dilation. At curriculum stage $t$, each ground-truth mask $\mathcal{Q}$ is dilated within $\mathcal{P}$ according to the following:

$$\widehat{\mathcal{Q}}_{\delta_t} = \{\, x \in \mathcal{P} \mid \min_{y \in \mathcal{Q}} \|x - y\|_2 < \delta_t \}, \quad \delta_t = \delta_0 \, \beta^{\lfloor t/\tau \rfloor}. \tag{2}$$

Here $\delta_0$ is the initial dilation radius, $\beta$ is the dilation factor, and $\tau$ is the step length for the dilation factor update.

Subsequently, for each predicted affordance element $j$, we construct its geometry descriptor $\mathcal{G}_j$, which captures the element's spatial properties, formally defined as $\mathbf{C}_j$ and $\mathbf{\Sigma}_j$ denote the position and size, respectively. We also employ a semantic descriptor $\mathbf{S}_j$, which, in conjunction with the geometric one, represents the affordance type for element $j$. In summary, these descriptors together form a compact yet visually unified representation of the 3D scene:

$$\mathcal{D}(\mathcal{P}) = \{(\mathbf{C}_j \in \mathbb{R}^3, \mathbf{\Sigma}_j \in \mathbb{R}^3, S_j)\}_{j=1}^N, \tag{3}$$

where $N$ is the total number of predicted affordance elements in the scene $\mathcal{P}$. These descriptors, capturing both geometric and semantic information of the affordance elements, provide a structured representation of the scene that facilitates subsequent reasoning.

**3D-2D Associations.** Using the generated surround-view images $\mathcal{V}$, we ground 3D affordance elements in these 2D views. For the predicted 3D elements with their descriptors $\mathcal{D}$, we project their 3D geometry onto every view $V_i \in \mathcal{V}$.

Specifically, we compute the element's 2D bounding box projection based on its 3D position and dimensions, assigning each projected box both a unique identifier linking it to the original 3D element $j$ and its corresponding affordance type $\mathbf{S}_j$ mapped to the 2D region. This process is formalized as:

$$\hat{V}_i = \mathcal{M}_{\text{3D} \to \text{2D}}\big(\mathcal{D}(\mathcal{P}), V_i\big), \tag{4}$$

where $\mathcal{M}_{\text{3D} \to \text{2D}}$ denotes our projection operator. This process effectively transfers key 3D information onto the 2D images.

Furthermore, to ensure legible MLLM input, we introduce an adaptive-labeling refinement strategy that resolves label collisions. This involves pre-defining candidate anchor positions around each projected box. When annotating an element, our pipeline iterates through these anchors, evaluating nonoverlap with existing elements, and selects the first suitable location. Such a lightweight spatial check effectively prevents label stacking, maintains clear object visibility, and provides an uncluttered canvas for subsequent MLLM reasoning.

Together, this collaborative representation enables downstream modules to access both the fine-grained geometry of affordance elements and their visual context within the scene, laying the groundwork for subsequent reasoning.

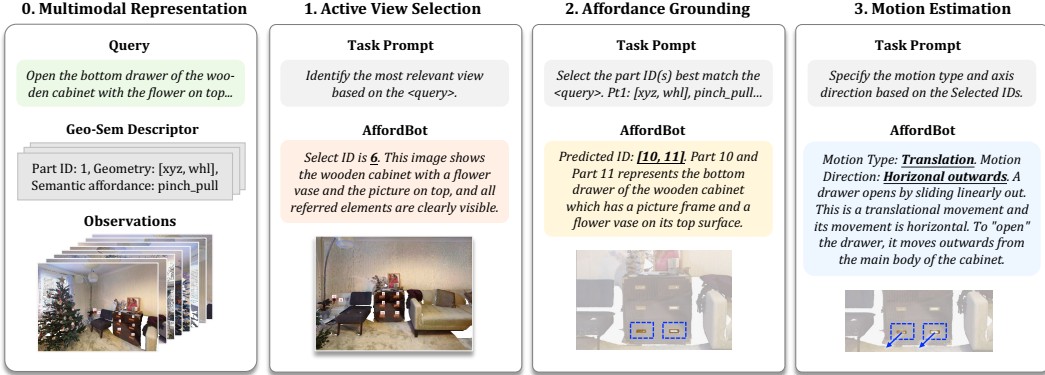

Figure 4: **AffordBot's Chain-of-Thought Pipeline for Embodied Reasoning.** This structured observe-then-infer process leverages multimodal inputs to perform: (1) Active View Selection to identify the most informative view, which may involve zooming in to better see the details of the images, followed by (2) Affordance Grounding to localize target elements, and finally (3) Motion Estimation to infer the required action details.

## 3.3 Chain-of-thought Reasoning

This section presents our method for enabling MLLM to perform embodied reasoning. Specifically, we design a tailored chain-of-thought paradigm that follows a structured observe-then-infer pipeline, as shown in Fig. 4. Grounded in the real physical world, our pipeline leverages visual observations to guide the MLLM through a sequence of inference steps: first actively selecting the most informative viewpoint (*observe*), followed by identifying targets and their required interactions in context (*infer*).

**Step 1: Active View Selection.** The goal of this step is to select the most informative view from the generated surround-view images $\hat{\mathcal{V}}$. Unlike prior methods that rely on instruction parsing and heuristic filtering over pre-processed features, we leverage the multimodal reasoning capabilities of an MLLM to guide view selection directly, enhancing both flexibility and accuracy.

Given the set of annotated views $\{\hat{V}_1, \ldots, \hat{V}_N\}$, where each includes projected 3D elements with unique IDs and affordance types, along with the instruction $\mathcal{T}$, the MLLM receives these as inputs. It is then tasked with selecting the view in which all referenced elements are visible and their identifiers are clearly shown, providing the most relevant visual content. The model directs attention to the selected view $\hat{V}_{\text{selected}}$, which serves as the semantically grounded visual input for subsequent reasoning.

**Step 2: Affordance Grounding.** This step aims to localize the specific affordance elements referenced by the linguistic instruction within the scene. Using the selected view $\hat{V}_{\text{selected}}$ from the previous step, the MLLM receives this annotated view, the original instruction $\mathcal{T}$, and detailed descriptors of the 3D affordance elements, including their unique IDs and spatial attributes. The MLLM interprets the instruction and visual cues to identify the region that best matches the described task. This process yields unique IDs of the localized target elements, which serve as a critical intermediate decision for the subsequent motion inference stage.

**Step 3: Motion Estimation.** This final step focuses on inferring the motion details of the elements localized in Step 2. The MLLM is provided with the original instruction $\mathcal{T}$, the image $\hat{V}_{\text{selected}}$, and its localized information from the previous step. Using these inputs, the MLLM deduces the intended action for the target element, including both the motion type and the direction of the motion axis.

To enable compatibility between continuous direction vectors and MLLM outputs, we discretize the former into interpretable categories. These categories broadly distinguish motion directions as horizontal or vertical, with specific refinements for translational movements (*e.g.*, inward/outward relative to the object's centroid) and rotational axes. This structured discretization ensures physical plausibility while remaining interpretable by the MLLM. The final output is an affordance-motion tuple that integrates the discretized motion representation with the model's language-driven reasoning.

Table 1: **Quantitative comparisons of our fine-grained embodied reasoning task.** We report the quantitative results of affordance grounding and motion estimation task on SceneFun3D [34] dataset.

| Task | | Grounding | | | | Motion | |
|---|---|---|---|---|---|---|---|
| Method | Raw 2D Input | mIoU | AP | $AP_{50}$ | $AP_{25}$ | +T | +TD |
| OpenMask3D [36] | ✓ | - | - | 0.0 | 0.0 | - | - |
| LERF [35] | ✓ | - | - | 4.9 | 11.3 | - | - |
| OpenMask3D-F [34] | ✓ | - | - | 8.0 | 17.5 | - | - |
| OpenIns3D [66] | ✗ | 0.0 | 0.0 | 0.0 | 0.0 | - | - |
| Fun3DU [37] | ✓ | 11.5 | 6.1 | 12.6 | 23.1 | - | - |
| Fun3DU (+motion) | ✓ | 10.0 | 4.6 | 9.9 | 18.7 | 11.5 | 4.0 |
| **AffordBot** | ✗ | **14.0** | **15.5** | **20.0** | **23.3** | **18.3** | **10.8** |

## 4 Experiments

This section presents a comprehensive experimental evaluation of our proposed fine-grained embodied reasoning framework, validating its effectiveness in joint affordance grounding and motion estimation. In addition, we conduct an in-depth analysis to assess how key module optimizations impact the system's overall accuracy and robustness.

### 4.1 Experimental Setup

**Dataset.** We conduct experiments on SceneFun3D [34], currently the only dataset that provides comprehensive annotations for fine-grained affordance grounding and motion estimation in 3D indoor scenes. It comprises a total of 230 richly annotated scenes, including 200 scenes for training, 30 for validation. Each scene provides dense point clouds annotated with element-level affordance masks, motion types, and motion axis directions. To facilitate instruction-oriented embodied reasoning for our task, we curate the annotation with task-specific annotation triples.

**Evaluation Metrics.** We adopt standard evaluation metrics [34] to assess performance on 3D affordance grounding and motion estimation. Specifically, we report mean Intersection-over-Union (mIoU), mean average precision (mAP), and average precision at IoU thresholds (AP, $AP_{25}$, $AP_{50}$), between predicted and ground-truth masks. To incorporate motion parameter accuracy, we adapt the $AP_{25}$ metric as proposed in [34, 51, 50], extending it with additional constraints on motion type and direction. Specifically, we further constrain mask prediction based on whether the model correctly predicts the motion type (+T), and both the motion type and motion axis direction (+TD).

**Implementation Details.** For visual-language reasoning, we employ Qwen2.5-VL-72B [15] locally deployed on four NVIDIA A800 GPUs. To construct the geometry descriptors and get segmented elements masks, we fine-tune Mask3D [65] from a pretrained checkpoint on ScanNet200 [67]. We train for 1,000 epochs on an NVIDIA A800 with the learning rate of 0.0001, a batch size of 2, and 2cm voxelization to preserve spatial detail.

### 4.2 Quantitative Results

The quantitative results for the fine-grained embodied reasoning on SceneFun3D dataset, as presented in Table 1, demonstrate the effectiveness of our AffordBot approach. By encoding 3D scenes into structured representations and processing them with task instructions via the MLLM, AffordBot outperforms existing methods including OpenMask3D [36, 34], LERF [35], OpenIns3D [66], and Fun3DU [37], as well as our enhanced Fun3DU (+motion) baseline. As shown in the table, AffordBot reports higher scores in both affordance grounding and motion estimation.

Notably, the results of our reproduced Fun3DU (+motion) baseline (second to last row) highlight the impact of incorporating a motion estimation branch into their original affordance grounding framework. For this baseline, we prompted Molmo [46] to infer motion parameters based on 2D segmentation results of affordance elements. The significant outperformance of AffordBot across all reported metrics underscores the advantage of our approach in accurately identifying and understanding the potential motions associated with affordance elements in 3D scenes. Specifically, our higher AP score, which is the average precision over IoU thresholds ranging from 0.5 to 0.95,

Table 2: **Ablation on key components of our Afford-Bot.** ALR denotes Adaptive Label Refinement, EVS denotes Enriched Visual Synthesis, and AVS denotes Active View Selection. Each variant incrementally incorporates one module, and finally Ex4 corresponds to our AffordBot.

|  | ALR | EVS | AVS | AP | $AP_{50}$ | $AP_{25}$ |
|---|---|---|---|---|---|---|
| Ex1 |  |  |  | 9.7 | 12.8 | 15.7 |
| Ex2 | ✓ |  |  | 9.7 | 13.0 | 16.1 |
| Ex3 | ✓ | ✓ |  | 14.8 | 19.4 | 22.1 |
| Ex4 | ✓ | ✓ | ✓ | **15.5** | **20.0** | **23.3** |

Table 3: **Ablation on viewpoint selection.** 'BEV' projects bird-eye view; 'Video Frame' uniformly samples frames from dataset, while 'Query-Aligned' picks the query-matching view; 'Ours' renders surround views for MLLM selection.

| Method | AP | $AP_{50}$ | $AP_{25}$ |
|---|---|---|---|
| BEV | 6.1 | 9.1 | 12.7 |
| Video Frame | 9.4 | 11.4 | 15.6 |
| Query-Aligned | 9.7 | 13.0 | 16.1 |
| Ours | **15.5** | **20.0** | **23.3** |

indicates that AffordBot not only performs well in rough localization but also maintains high precision under stricter localization requirements (*i.e.*, higher IoU thresholds). This is crucial for robotic manipulation tasks, where precise segmentation is essential for accurate grasping and manipulation. Furthermore, the results suggest the importance of grounding accuracy for subsequent tasks and indicate the enhanced spatial awareness provided by 3D-based motion reasoning.

### 4.3 Ablation Studies

To quantify the contribution of AffordBot, we conduct ablation studies focusing solely on the affordance-grounding task, which is the critical prerequisite for motion estimation and downstream execution. This targeted approach is justified because the investigated modules (representation design and the MLLM-driven view-selection mechanism) operate entirely upstream of motion estimation. Once a target is accurately grounded, motion prediction relies solely on that grounded object and a fixed MLLM prompt. Consequently, any modifications to these upstream components propagate through the entire pipeline and are comprehensively reflected in grounding performance metrics.

**Ablation on Key Components.** Through systematic component-wise analysis, we demonstrate how progressive module integration contributes to the performance, as shown in Tab. 2. Baseline Ex1, adapted from [68], initially employs the MLLM to parse instructions and identify target affordance types. It then renders all matching segmented elements from the scene's center view, annotating each with unique identifiers at their 2D centroids. Finally, the MLLM processes these rendered views to localize the correct element. Adding Adaptive Label Refinement (ALR, Ex2) identifier labels to avoid occlusion, our method yields a modest but consistent lift of $+0.4\%$ $AP_{25}$. The major improvement comes from enriched visual synthesis (EVS, Ex3). The model gains much richer context, pushing $AP_{25}$ from 16.1% to 22.1% ($+6.0\%$). This significant improvement demonstrates that global, information-dense observation, as provided by EVS, is much more valuable than the single frame used in [68]. Finally, Ex4 employs the active view selector (AVS). This *focus-then-infer* pipeline both trims redundant visual information and exploits the best evidence, raising $AP_{25}$ to 23.3% and achieving the highest overall accuracy.

**Ablation on Viewpoint selection.** To probe how viewpoint choice affects subsequent inference, we evaluate this in Tab. 3. While Bird's-Eye View (BEV) representations provide scene overview, they prove ineffective for our task as affordance elements typically require fine-grained appearance details due to their small size. Sampling images directly from the video stream (*i.e.* Video Frame baseline) also yields limited effectiveness, outperforming the previous results. Query-aligned baseline retrieves a single pre-tagged frame whose affordance class matches the query.

Our Dynamic strategy takes a different route: it first synthesises a dense 360° sweep of surround views, then asks the MLLM to select the frame that best matches the instruction. This active "observe-then-infer" routine supplies rich global context while keeping the final input compact, boosting $AP_{25}$ to 23.3%, an absolute gain of 7.6% over the query-aligned method, as shown in Tab. 3.

**Probing the Primary Bottleneck of AffordBot.** To investigate the primary bottlenecks constraining upstream segmentation and downstream active view selection performance, we progressively replace Mask3D's predicted masks with ground-truth masks (GT proposals) and provide an ideal front-view perspective (GT proposals + views), as summarized in Table 4. The first row shows the baseline configuration (Mask3D proposals), which corresponds to our AffordBot. Replacing the predicted

Table 4: **Probing the bottleneck of our method**. 'Mask3D proposals' refers to our Affordbot, which uses predicted proposals. 'GT proposals' denotes the use of ground-truth masks, while 'GT proposals & views' additionally adopt ground-truth views.

| Method | AP | AP$_{50}$ | AP$_{25}$ |
|---|---|---|---|
| Mask3D proposals | 15.5 | 20.0 | 23.3 |
| GT proposals | 35.7 | 39.4 | 45.4 |
| GT proposals & views | 38.3 | 42.3 | 47.4 |

Table 5: **Comparison of different MLLMs.** Deployable models (LLaVA-v1.6-34B, Qwen2.5-VL-72B) and commercial GPT APIs show consistent trends, with larger models yielding stronger performance.

| Method | AP | AP$_{50}$ | AP$_{25}$ |
|---|---|---|---|
| LLaVA-v1.6-34B | 10.6 | 14.2 | 16.9 |
| Qwen2.5-VL-72B | 15.5 | 20.0 | 23.3 |
| GPT-4o | 16.5 | 22.1 | 28.9 |
| GPT-o1 | 24.8 | 30.3 | 33.4 |

Table 6: **Performance variation across affordance types**. 'Segment' measures upstream segmentation accuracy, while 'Reason' reports final grounding.

| Type | rotate | key_press | tip_push | hook_pull | pinch_pull | hook_turn | foot_push | plug_in | unplug |
|---|---|---|---|---|---|---|---|---|---|
| Segment | 0.0 | 11.3 | 5.3 | 27.5 | 27.1 | 67.8 | 100.0 | 11.1 | 15.3 |
| Reason | 2.5 | 30.4 | 5.1 | 18.0 | 23.5 | 45.1 | 100.0 | 8.3 | 16.7 |

Table 7: **Performance variation with different numbers of target elements**. We compare tasks involving a single ground-truth element ('Unique') *versus* multiple elements ('Multiple').

| Method | mIoU | AP | AP$_{50}$ | AP$_{25}$ | +T | +TD |
|---|---|---|---|---|---|---|
| Unique | 13.2 | 13.8 | 18.2 | 21.4 | 16.5 | 9.9 |
| Multiple | 19.1 | 27.2 | 32.1 | 35.8 | 30.2 | 17.0 |
| Overall | 14.0 | 15.5 | 20.0 | 23.3 | 18.3 | 10.8 |

masks with GT proposals results in a 22.1% boost in AP25, highlighting that *instance segmentation noise is the primary limiting factor*. With perfect segmentation, adding the optimal viewpoint further improves performance by +2.0% AP25, suggesting that while active perception can still be optimized, it is not the dominant bottleneck.

**Comparison of Different MLLMs.** We replace the default Qwen2.5-VL-72B with several representative alternatives (LLaVA-v1.6-34B, GPT-4o, and GPT-o1), as reported in Tab. 5. While the commonly used Qwen achieves 23.3% AP$_{25}$, adopting the more advanced GPT-o1, which features superior reasoning and visual understanding, further boosts performance to 33.4% AP$_{25}$. This demonstrates that leveraging stronger MLLMs can unlock even greater potential within our framework.

**Performance Variation Analysis.** We conduct a detailed performance analysis to uncover variations across different affordance types and target element counts. Tab. 6 highlights significant disparities in AP$_{50}$ across different affordance types, partly reflecting dataset class imbalance and a strong dependence on initial segmentation quality. Performance ranges widely (*e.g.*, 100% for "foot_push" vs. 0% for "rotate"]), limiting grounding accuracy. The MLLM improves performance for some categories using linguistic cues (*e.g.*, "key_press"), but challenges remain in aligning visual and language cues for others (*e.g.*, "tip_push" degradation).

Further analysis of task subsets (Tab. 7) reveals better performance for "Multiple" target elements than "Unique" ones. This difference is primarily due to Mask3D struggling with the small, weakly-textured objects typical of "Unique" instances, resulting in noisier descriptors that hinder subsequent affordance grounding and motion estimation.

## 4.4 Qualitative Results

Fig. 5 provides detailed qualitative results of our fine-grained reasoning task. AffordBot generally shows more accurate and consistent grounding of target elements, particularly in complex scenarios with multiple or small targets. Overall, qualitative evidence further suggests that AffordBot achieves significantly improved affordance understanding compared to the prior SOTA method [37].

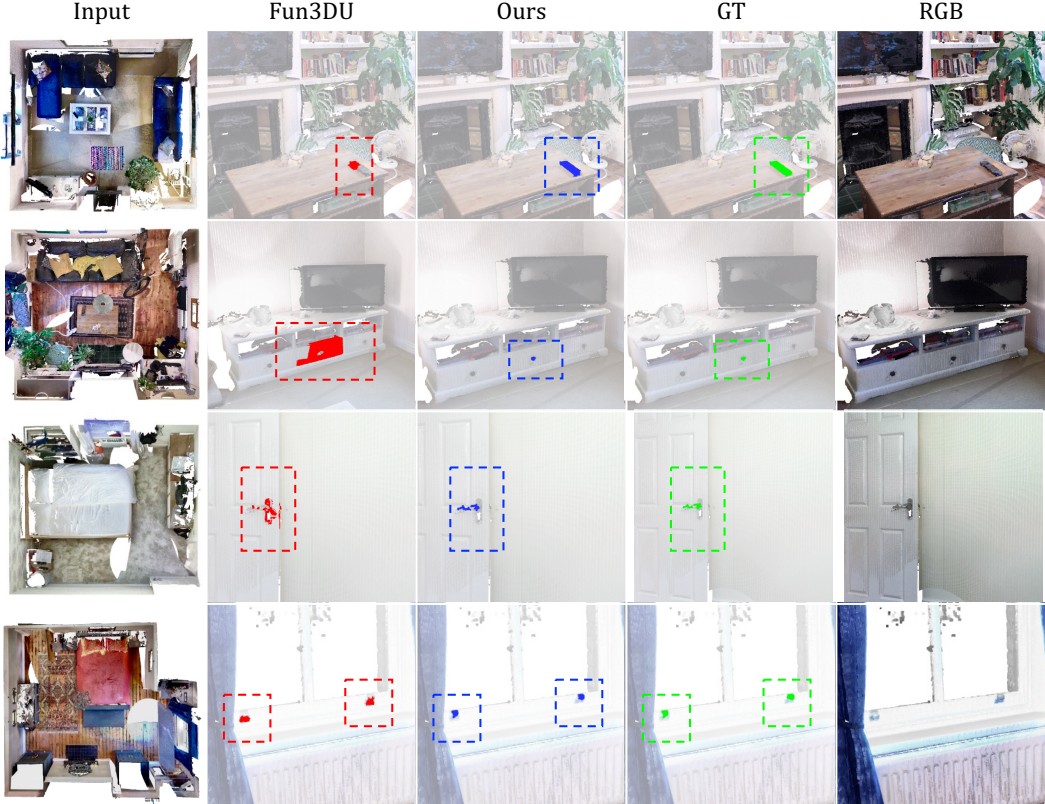

| Input | Fun3DU | Ours | GT | RGB |
|---|---|---|---|---|

Figure 5: **Qualitative Results.** The figure showcases visual examples of AffordBot performing fine-grained grounding. The illustrated examples include: (1) *"Turn on the TV using the remote control on the table."* (2) *"Open the middle drawer of the TV stand."* (3) *"Close the bedroom door."* (4) *"Open the window above the radiator"*. Please zoom in digitally to view more details.

## 5  Conclusion

Fine-grained embodied reasoning in 3D worlds serves as a crucial bridge from perception to action, making it pivotal for agents to perform sophisticated tasks. While prior work has explored affordance grounding and motion estimation in isolation, our work unifies these tasks under a structured reasoning framework, AffordBot, bridging perception and action through instruction-aware triplet prediction. By leveraging MLLM with a tailored chain-of-thought paradigm, our method ensures physically grounded reasoning that advances from scene perception to affordance localization and motion synthesis. Through extensive experiments, AffordBot not only advances state-of-the-art performance but also demonstrates the feasibility of efficient, task-coherent embodied reasoning, paving the way for more intuitive human-agent interaction in complex 3D spaces. In the future, we will empower AffordBot with more advanced multimodal understanding ability [69].

## Acknowledgments

This work was supported by the National Natural Science Foundation of China (NSFC) under Grant U22A2094, and also supported by the Ministry of Education, Singapore, under its MOE Academic Research Fund Tier 2 (MOE-T2EP20124-0013). We also acknowledge the support of the Supercomputing Center of USTC for providing advanced computing resources and of the NSFC with Grant No. 62573371.

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
