# OpenReview forum: "AffordBot: 3D Fine-grained Embodied Reasoning via Multimodal Large Language Models"
_NeurIPS.cc/2025/Conference — NeurIPS 2025 poster_

### Official Review · Reviewer_HoJn · 2025-06-23

**Clarity:** 3
**Significance:** 2
**Originality:** 2
**Rating:** 4
**Confidence:** 5

**Summary:**

The paper proposes AffordBot, a framework for fine-grained 3D embodied reasoning. Given 3D point clouds and task instructions, it predicts affordance masks, motion types, and directions for target elements. The approach projects 3D data into surround-view images, uses MLLMs with chain-of-thought reasoning for active view selection, affordance grounding and motion estimation, and demonstrates state-of-the-art results on SceneFun3D.

**Questions:**

1. The current framework relies on 2D MLLMs after projecting 3D point clouds into 2D views. Have the authors considered leveraging recent 3D-aware MLLMs or directly incorporating point cloud features into the reasoning pipeline? Could the method benefit from native 3D spatial reasoning rather than relying on 2D projections?
2. The approach appears highly dependent on the initial segmentation quality from Mask3D. Could the authors provide more analysis or ablations on how segmentation noise affects downstream reasoning performance? How robust is AffordBot when upstream segmentation contains errors or uncertainties?
3. The evaluation is limited to SceneFun3D. Could the authors test on additional datasets or real-world scans to verify the generalization ability of the framework? How would the method handle out-of-distribution instructions or affordances not seen in SceneFun3D?

**Ethical Concerns:**

["NO or VERY MINOR ethics concerns only"]

**Final Justification:**

I thank the authors for their rebuttal and clarifications. Most of my concerns have been addressed. After considering the response and the opinions of the other reviewers, I have decided to raise my original score to 4.

**Limitations:**

Yes

**Paper Formatting Concerns:**

No major formatting issues.

**Quality:**

2

**Strengths And Weaknesses:**

Strengths:
1. The paper addresses the problem of fine-grained embodied reasoning in 3D scenes, combining affordance grounding and motion estimation in a unified task.
2. The paper demonstrates superior performance over existing baselines on the SceneFun3D dataset, with comprehensive quantitative, ablation, and qualitative analyses.
3. The paper is well-organized and easy to follow.

Weaknesses:
1. The proposed approach avoids the core challenges of 3D reasoning by projecting 3D point clouds into 2D surround views and relying on 2D MLLMs for reasoning. While effective, this design overlooks the need for deeper spatial understanding and physical reasoning in 3D space, which are central to embodied reasoning. Moreover, recent advances have explored directly incorporating point cloud inputs into MLLMs or 3D-aware vision-language models, but this direction is not sufficiently explored.
2. As indicated in Table 4 and 5, the method’s performance is heavily dependent on the initial Mask3D segmentation quality. The approach may struggle with small or occluded objects where segmentation is less reliable, leading to potential failures in downstream reasoning steps.
3. The evaluation focuses solely on SceneFun3D, which is a single-domain dataset. The generalization ability to more diverse real-world environments, or more complex instructions remains untested. Additional benchmarks would validate the method’s generalization applicability.

---

> ### Author Rebuttal · Authors · 2025-07-31
>
> Thank you for your valuable comments. We appreciate your positive assessment of our method’s performance and analysis. We hope the responses below address your concerns.
>
> **W1.1: Clarification on 3D-aware reasoning.**
>
> We sincerely appreciate your recognition of our method's effectiveness. However, we respectfully disagree with the concern that our design overlooks deeper 3D spatial or physical reasoning. On the contrary, it preserves 3D spatial information by explicitly associating each element in the 2D rendered views with its precise 3D coordinates. These associations are incorporated into the MLLM input as reasoning-aware cues—each image is overlaid with bounding boxes and paired with textual prompts that enumerate the corresponding 3D attributes. As a result, the model operates not on raw 2D pixels alone, but on spatially grounded visual–textual inputs, enabling effective 3D-aware reasoning throughout the chain-of-thought process.
>
> **W1.2&Q1: Comparison to native 3D MLLMs.**
>
> Thanks for your valuable comments. The cutting‑edge 2D MLLMs are currently an order of magnitude larger and more capable than 3D‑aware models in both vision-language understanding and long-horizon reasoning, as evidenced by our superior performance.  While leveraging 3D-aware MLLMs or directly incorporating point cloud features is promising, such directions currently face several limitations. First, our task requires fine-grained reasoning on small affordances (e.g., knobs, handles) embedded in cluttered 3D scenes, which is particularly challenging. However, most existing 3D MLLMs operate at a coarse object-level granularity and struggle to support our fine-grained task. Moreover, incorporating point cloud features into MLLMs would require architectural modifications and retraining. These scene-level architectures typically require large-scale training data, for instance, 3D-LLM [1] utilizes 300K 3D-text pairs, whereas our current dataset SceneFun3D (230 scenes and 4K instructions) remains quite limited.
>
>
> **W2&Q2: Dependence on segmentation quality.**
>
> We appreciate the reviewer’s feedback regarding the dependence on Mask3D segmentation. As acknowledged in Section 4.2 (Lines 290–292), our framework does rely on upstream segmentation to propose candidate regions. However, the subsequent multimodal representation and chain-of-thought reasoning provide a level of robustness that can mitigate some of these errors. This is evident in Table 4, where categories such as "rotate" and "key_press"—despite lower segmentation quality—still yield strong downstream performance. Moreover, compared to the prior state-of-the-art Fun3DU, which also adopts a multi-stage architecture, our method achieves substantially higher AP50 performance, suggesting stronger tolerance to segmentation noise.
> We also agree that small or occluded affordance elements pose an even greater challenge for segmentation. We plan to explore hybrid segmentation strategies that combine 2D instance cues with 3D geometry, or leverage strong 2D segmentation priors as cross-modal constraints to better localize hard-to-see instances.
> To better quantify the impact of segmentation quality, we conducted two analyses:
>
> (i) Performance upper bound: We replaced the predicted instance masks with ground-truth segmentation. As shown below, AP50 improves from 20.0 to 39.4, indicating significant headroom and confirming segmentation as a key bottleneck.
>
> |   Upstream Method   | mAP  | AP50 | AP25 |
> |---------------------|------|------|------|
> | Predicted proposals | 15.5 | 20.0 | 23.3 |
> |   GT proposals      | 35.7 | 39.4 | 45.4 |
>
> (ii) Robustness under noise:
> We simulate segmentation errors by randomly deleting a percentage of predicted masks. Interestingly, the downstream performance remains stable up to 20% noise. The performance only begins to decline more noticeably when deletion reaches 30%:
>
> | Noise Ratio | mAP  | AP50 | AP25 |
> |-------------|------|------|------|
> |     0%      | 15.5 | 20.0 | 23.3 |
> |     10%     | 15.4 | 20.0 | 23.8 |
> |     20%     | 16.4 | 19.8 | 22.6 |
> |     30%     | 13.3 | 17.5 | 21.9 |
> |     40%     | 11.7 | 15.1 | 18.0 |
>
> These results reflect the strength of the reasoning module, which is capable of partially recovering from upstream failures. This finding outlines a promising direction for future work: employ more robust segmentation strategies or leverage the multimodal reasoning capabilities of subsequent MLLMs to iteratively refine segmentation outputs. We will incorporate these findings into the revised version.
>
>
> **W3&Q3: Additional evaluation of generalization ability.**
>
> We appreciate the reviewer’s suggestion regarding the investigation of generalization capability. However, to the best of our knowledge, SceneFun3D is currently the only available scene-level dataset for our task. Preparing new 3D datasets with compatible labels requires extensive manual annotation and is unfortunately infeasible within the rebuttal period. To address the generalization concern, we conducted a test using GPT-rephrased SceneFun3D instructions with more diverse syntax and complex phrasing to evaluate the model’s robustness to out-of-distribution language inputs. Our method maintains comparable performance across original and rephrased instructions, with only a minor drop. In contrast, the baseline Fun3DU suffers a significant performance drop. These results suggest that AffordBot generalizes well to more diverse and complex instruction formulations, validating its robustness to out-of-domain language.
>
> |       Method       | mAP  | AP50 | AP25 |
> | ------------------ | ---- | ---- | ---- |
> | Fun3DU (original)  | 6.1  | 12.6 | 23.1 |
> | Fun3DU (rephrased) | 3.7  | 8.1  | 15.1 |
> | Ours (original)    | 15.5 | 20.0 | 23.3 |
> | Ours (rephrased)   | 13.7 | 18.5 | 22.6 |
>
> **References:**
>
> [1] Hong Y, Zhen H, Chen P, et al. 3d-llm: Injecting the 3d world into large language models[J]. Advances in Neural Information Processing Systems, 2023, 36: 20482-20494.

---

> > ### Comment · Area_Chair_kpj9 · 2025-08-07
> >
> > Dear Reviewer HoJn,
> >
> > This is the only negative recommendation among the reviewers. Could you please check whether the rebuttal adequately addresses your concerns? Thanks for your efforts.
> >
> > Best regards,
> >
> > Your AC

---

> ### Author Response · Authors · 2025-08-06
> **Follow-up on Reviewer Feedback**
>
> Dear Reviewer HoJn,
>
> Thank you for your insightful comments and thoughtful suggestions, which have been invaluable in improving our paper.
>
> As the author-reviewer discussion period nears its end, we would greatly appreciate it if you could kindly confirm whether our responses have adequately addressed your concerns.
>
> Thank you once again for your time and invaluable guidance.
>
> Best regards,
> The Authors

---

> > ### Comment · Reviewer_HoJn · 2025-08-07
> >
> > I thank the authors for their rebuttal and clarifications. Most of my concerns have been addressed. After considering the response and the opinions of the other reviewers, I have decided to raise my original score to 4.

---

> > > ### Author Response · Authors · 2025-08-07
> > > **Thank you for your response!**
> > >
> > > Thank you for recognizing our efforts in addressing your concerns and for your decision to raise the score! We will incorporate your suggestions into the final version.

---

### Official Review · Reviewer_m6AC · 2025-06-30

**Clarity:** 4
**Significance:** 3
**Originality:** 2
**Rating:** 5
**Confidence:** 5

**Summary:**

This paper teaches an agent to jointly perform 3D affordance grounding and motion estimation based on a natural language instruction. For a given 3D scene and command, the agent must predict a structured triplet (3D mask, motion type, motion axis) for each relevant interactable part. To address this task, the authors propose AffordBot, a novel framework that leverages Multimodal Large Language Models (MLLMs). A key aspect of the framework is its method for bridging the 3D-to-2D modality gap by rendering surround-view images of the scene and projecting 3D element candidates onto them. AffordBot then employs a tailored Chain-of-Thought (CoT) reasoning paradigm, which notably begins with an "active perception" stage where the MLLM selects the most informative viewpoint before proceeding to localize the target element and infer its motion parameters. Experiments on the SceneFun3D dataset show that AffordBot achieves state-of-the-art performance.

**Questions:**

1.  The absolute performance, while state-of-the-art, remains relatively low (e.g., an overall AP of 15.5 and AP50 of 20.0). This suggests the task is extremely challenging. Could the authors provide a more in-depth analysis of the primary bottlenecks? Based on your experiments, what are the main sources of error? Is it the upstream segmentation quality (as hinted in Table 4), the MLLM's spatial reasoning capabilities, ambiguities in the language instructions, or the inherent difficulty of the dataset itself? A detailed error analysis would be highly valuable for future work in this area.

2.  The active view selection mechanism is a core contribution. Could you elaborate on its failure modes? Are there cases where the MLLM selects a suboptimal or incorrect view, and if so, how does the system's performance degrade? An analysis of the view selector's standalone accuracy and its downstream impact on the grounding and motion estimation tasks would provide deeper insight into its robustness.

3.  The experiments are conducted using the Qwen2.5-VL-72B model. How dependent are the framework and the CoT paradigm on this specific MLLM? Have you experimented with other MLLMs (e.g., LLaVA, OpenAI o-series)? It would be helpful to understand whether the success is primarily due to the power of this specific model or the generality of the AffordBot framework itself.

4. As far as I see, the advantage that a vision-language model brings in this task is the possibility to conduct long-horizon reasoning in order to finish a certain goal. Have you tried the capability of AffordBot in such settings like [1]?

[1] PreAfford: Universal Affordance-Based Pre-Grasping for Diverse Objects and Environments

**Ethical Concerns:**

["NO or VERY MINOR ethics concerns only"]

**Final Justification:**

See my comments below.

**Limitations:**

Please see the weakness section.

**Quality:**

3

**Strengths And Weaknesses:**

Strengths

1/ Effective Framework Design: The AffordBot framework is well-conceived and effectively tackles key challenges. The method for creating a rich multimodal representation from a 3D point cloud—by synthesizing surround-views and projecting 3D geometric-semantic descriptors—is a smart solution to make 3D scenes accessible to 2D-native MLLMs without relying on computationally expensive video processing.

2/ Thorough Experimental Validation: The paper provides a comprehensive set of experiments that convincingly validate the proposed approach. The overall results demonstrate a clear state-of-the-art performance. The component-wise ablation studies are particularly strong, systematically quantifying the contribution of each module (ALR, EVS, AVS) and providing clear evidence for the authors' design choices.

Weaknesses

1/ Heavy Reliance on Upstream Segmentation: The performance of the entire AffordBot pipeline is critically dependent on the quality of the initial 3D instance segmentation from the Mask3D model. The authors' own analysis in Table 4 shows that poor segmentation accuracy for certain affordance types directly leads to low final grounding accuracy. This indicates that errors from the upstream module can propagate and are not recoverable, making the system brittle. The framework is not end-to-end, which may limit its generalization ability in real-world scenarios with imperfect perception.

2/ Impact of Motion Discretization: The method discretizes continuous motion axis directions into a set of text-based categories to be compatible with the MLLM's output format. While this is a practical solution, it is a simplification that inherently loses precision. The paper does not discuss the potential impact of this information loss or analyze the error introduced by this categorization scheme.

---

> ### Author Rebuttal · Authors · 2025-07-31
>
> Thank you for your valuable comments. We are pleased that the reviewer recognizes our method as novel, effective, and thoroughly evaluated. Please see our responses below.
>
> **W1: Reliance on upstream segmentation.**
>
> Thank you for the valuable insights. We agree that the quality of upstream segmentation impacts overall performance. Nonetheless, our two‑stage design already achieves the strongest results. Specifically, AffordBot attains 20.0 AP50 on SceneFun3D, which significantly outperforms the previous SOTA Fun3DU (12.6 AP50). While an end-to-end approach would be elegant theoretically, such alternatives require large-scale training. Given the current dataset scale (230 scenes and 2K instructions), our solution remains the optimal choice. The data scaling and the exploration of end-to-end architectures should be left as future work. We will incorporate these analyses in our revision.
>
> **W2: Impact of motion discretization.**
>
> We thank the reviewer for the thoughtful feedback. In practice, the vast majority of motion-related objects in SceneFun3D—such as drawers, doors, sockets, and switches—are intentionally aligned along the primary axes, reflecting their real-world physical constraints. Therefore, discretizing the motion axis into a small number of semantic categories is not merely a simplification, but a design choice grounded in the spatial regularity of common indoor affordances. Moreover, the predicted motion axis in our framework serves as a high-level reasoning output in the execution system—analogous to a planning signal that instructs "pull outwards horizontally." This allows subsequent low-level modules or agents to refine and execute the motion with more precision. In this sense, the discretization operates as an abstraction appropriate for reasoning, not for direct actuation. To quantify potential information loss, we analyzed the angular deviation between ground-truth motion axes and their nearest canonical direction. We found that over 58% of all instances fall within 10° of the nearest primary axis, and over 96% fall within 30°. This reflects that most motion directions in our dataset naturally concentrate around the canonical axes we designed.
>
> **Q1: In-depth analysis of bottlenecks.**
>
> Thank you for your inspiring suggestion! We agree that the overall affordance grounding performance is still relatively low. This reflects the large difficulty of the SceneFun3D dataset and our task. It elevates 3D affordance reasoning to the scale of entire scenes, requiring a system to parse millions of points in indoor environments guided by natural-language instructions to precisely localize small affordance elements such as "knob" or "handle", which obviously showcase the challenging nature of the benchmark. Under such conditions, our approach still improves the previous state-of-the-art AP50 score from 12.6 to 20.0, nearly doubling it, demonstrating that our AffordBot delivers meaningful progress even in such a demanding setting.
> To dissect the primary bottlenecks, we progressively replaced Mask3D’s predicted masks with ground-truth (GT) masks and further provided an ideal front-view perspective. The results reveal that correcting segmentation alone boosts AP50 by nearly 20 percentage points, indicating that instance segmentation error is currently the primary limiting factor. With perfect segmentation, further providing an optimal viewpoint yields an additional improvement of 2.9 pp, suggesting that active perception still has room for optimization but is not the major bottleneck. Notably, when given both perfect masks and viewpoints, AP25 only reaches 47.4, exposing an additional challenge in cross-modal geometric reasoning—MLLMs may be confused about target localization in some multi-instance or complex spatial relation scenarios. We will add these in-depth analyses in the revised paper, hoping that our method and its thorough dissection will further inspire future work in the community.
>
> |         Setting         | mAP  | AP50 | AP25 |
> |-------------------------|------|------|------|
> | Mask3D proposals        | 15.5 | 20.0 | 23.3 |
> | GT proposals            | 35.7 | 39.4 | 45.4 |
> | GT proposals & GT views | 38.3 | 42.3 | 47.4 |
>
> **Q2: In-depth analysis of Active View Selection.**
>
> Thank you again for recognising the contribution of our Active View Selection (AVS) mechanism. AVS leverages MLLM to select the most informative viewpoint based on the observation, which enables the model to perform better. As reported in Table 3, replacing AVS with a simple fixed viewpoint obtained by uniformly sampling video frames lowers AP50 by 8.6 pp, underscoring AVS’s core contribution. Motivated by your insights, we analyzed the failure cases of AVS and found two main patterns: (i) the target part is occluded by objects further ahead, making it absent or blurry in the image; (ii) several instances of the same category and appearance coexist in the scene as distractors.
> Furthermore, to quantify AVS accuracy, we adopt a strict criterion: a selection is considered successful only if every ground‑truth target element is visible and unobstructed. Under this, AVS achieves a success rate of 56%, which reflects that although our current strategy design has significantly improved compared to previous methods, there is still room for improvement. We will include this analysis in the revised version and, for future work, plan to explore more robust strategies such as multi‑candidate view generation or post-refinement.
>
> **Q3: Additional experiments with different MLLMs.**
>
> The success of our AffordBot stems from its integrated pipeline, which comprises multimodal scene representation, active view selection, and chain‑of‑thought prompting. It can be easily integrated into different MLLMs and fully utilize their strong reasoning capabilities. To verify this, we replaced Qwen‑2.5‑VL‑72B with several representative alternatives: LLaVA‑v1.6‑34B and the commercial GPT series. These results demonstrate that AffordBot is not tied to a particular model and maintains strong performance across MLLMs. Moreover, the results suggest that the better the MLLM’s vision-language capability, the higher the overall grounding accuracy. We will supplement these findings in the revised paper.
>
> |   MLLM    | mAP  | AP50 | AP25 |
> |----------------|------|------|------|
> | LLaVA-v1.6-34B | 10.6 | 14.2 | 16.9 |
> | Qwen2.5‑VL‑72B | 15.5 | 20.0 | 23.3 |
> |     GPT-4o     | 16.5 | 22.1 | 28.9 |
> |     GPT-o1     | 24.8 | 30.3 | 33.4 |
>
> **Q4: MLLM capabilities and extended settings.**
>
> We fully agree with the reviewer that one of the most exciting promises of an MLLM‑driven pipeline is its capacity for truly long‑horizon reasoning. In our paper, however, we focus on the high-level reasoning as aligned with the existing benchmark. Extending AffordBot to a setting akin to PreAfford, where the agent plans and executes a chain of affordance‑conditioned actions that progressively reshape the environment, would be an appealing next step. Implementing this requires not only multi‑turn CoT prompting but also a closed‑loop simulation that updates the scene after each predicted action, so that subsequent reasoning can take the new state into account. Because such an end‑to‑end simulator integration and experimental campaign lie beyond the time available for the rebuttal period, we plan to explore this in future work.

---

> > ### Comment · Reviewer_m6AC · 2025-08-02
> > **Reply to rebuttal**
> >
> > Most of my major concerns are well addressed. Really appreciate the efforts the authors made during the rebuttal perioid. I will increase my score by 1. Hopefully the in-depth discussion introduced in the rebuttal period will be included in the final revision of the manuscript.

---

> > > ### Author Response · Authors · 2025-08-02
> > > **Thank you for your response!**
> > >
> > > Thank you for the update and positive feedback! We’re glad to hear your major concerns have been addressed and will incorporate the added analyses into the final version.

---

### Official Review · Reviewer_qb6v · 2025-07-01

**Clarity:** 2
**Significance:** 2
**Originality:** 2
**Rating:** 5
**Confidence:** 3

**Summary:**

This paper introduces AffordBot, a method that uses 2D MLLM for embodied reasoning in 3D scenes. It utilizes view synthesis and Geometry-Semantic Descriptors for scene representation, then propose to use MLLM for step-by-step affordance grounding by first select the most informative view and then ground the action label and axis. The resulting pipeline demonstrates improved performance on SceneFun3D dataset in terms of grounding accuracy and motion prediction accuracy. Extensitve ablation studies show the effectiveness of the proposed method.

**Questions:**

1. Have the authors tried to understand how the model would choose between views when the task can be applied to multiple instances?

2. In the abstract, the authors state: we introduce a new task: Fine-grained 3D Embodied Reasoning. How is this task different from SceneFun3D?

3. Have the authors tried to understand the effect of changing the number of views?

**Ethical Concerns:**

["NO or VERY MINOR ethics concerns only"]

**Final Justification:**

The author has mainly addressed my concerns and solved my confusion about the task setting w.r.t SceneFun3D. I'll increase my scroe to 5

**Limitations:**

See Questions and Weaknesses.

**Quality:**

3

**Strengths And Weaknesses:**

Strengths:

1. AffordBot proposes a scene representation and a reasoning chain that effectively help with better affordance and motion grounding in the 3D scene.

2. Extensive experiments and ablation studies show the effectiveness of the proposed method.

3. Analyses of the experimental results are comprehensive.

Weaknesses:

1. The proposed view synthesis method seems to be less effective when the structure of the house is complex, for examle, taking a 360 views at the center of house 0 in ScanNet will result in no observations of the restroom and some cabinets in the kitchen.

2. The proposed active view selection chose the most informative view for the downstream grounding, which could cause the issue of ambiguity when there're multiple objects of the same category.

3. To avoid the problems raised in Figure 3, the number of frames generated by the view synthesis stage would be huge to get a good coverage and enough context for the downstream tasks, which could be inefficient.

---

> ### Author Rebuttal · Authors · 2025-07-31
>
> Thank you for your valuable comments. We are pleased that the reviewer recognizes our method's effectiveness, thorough experiments, and comprehensive analyses. Please see our responses below.
>
> **W1: View‑coverage considerations in complex environments.**
>
> Thank you for this valuable insight. In our current experiments, we use the SceneFun3D dataset, where annotated affordance elements are densely distributed in the main room. Under this distribution, our surround‑view synthesis strategy can cover almost all labeled elements. We agree that in environments with complex layouts and multiple sub‑rooms, such as those found in ScanNet, a single panoramic sweep may fail to capture occluded areas like restrooms. While our current implementation demonstrates an initial step toward the fine-grained embodied reasoning task, we plan to extend this in future work by developing more comprehensive observation strategies, including incremental observation, to better handle complex scenes.
>
> **W2&Q1: Multi‑instance scenarios.**
>
> We thank the reviewer for the thoughtful insights. Indeed, in scenarios where multiple affordance instances match the instruction, the language input often includes referential cues (such as "open the top drawer of the right cabinet") to help disambiguate. In our current pipeline, we let the MLLM itself select the most informative view, based on the tailored prompt and the multimodal representation. This design allows the model to leverage spatial references in language to resolve ambiguity among identical objects. During the subsequent reasoning step, the MLLM is required to locate all instances that satisfy the instruction, which may include one or multiple valid instances. Even without explicit disambiguation modules, our method demonstrates stronger understanding than previous work.
>
> **W3&Q3: Analysis of view coverage and quantity.**
>
> We respectfully disagree. The images shown in Fig. 3 are original RGB frames from the dataset, not our synthesized views. Precisely to address the limited context in such original observations, our surround-view synthesis module (Figure 2, middle) generates comprehensive coverage using only 8 synthesized views, significantly reducing the number of frames required while mitigating the visibility issues shown in Figure 3. To understand the impact of view quantity, we test 2, 4, 8, and 16 views. As shown in the table below, even 4 views offer strong coverage and performance, while 8 views provide complete 360° scanning with peak accuracy. However, increasing to 16 views leads to performance degradation due to redundant overlaps and the limitations of our MLLM (Qwen2.5-VL-72 B), which supports at most 9 images—forcing us to tile views and reduce resolution. Given this circumstance, we adopt 8 views as the best trade-off between efficiency and accuracy and will include this analysis in the revised version.
>
> | No. of Views | mAP  | AP50 | AP25 |
> |--------------|------|------|------|
> |      2       | 11.5 | 14.7 | 18.5 |
> |      4       | 14.5 | 19.0 | 22.6 |
> |      8       | 15.5 | 20.0 | 23.3 |
> |      16      | 10.1 | 13.9 | 17.6 |
>
> **Q2: Difference from SceneFun3D tasks.**
>
> As outlined in Section 1, SceneFun3D provides multiple independent tasks (affordance segmentation, affordance grounding, and motion estimation) with separate benchmarks. However, this pipeline processes subtasks in isolation and predicts motions for all functional parts independently of user instructions, limiting its applicability in real-life scenarios. To overcome these, our task unifies affordance-mask localization, motion type, and motion axis into a structured triplet prediction, enabling joint spatial grounding and interaction reasoning in a single stage. Besides, our task estimates motion parameters only for instruction-relevant affordances, while SceneFun3D operates in an instruction-agnostic manner. These differences in task formulation and output granularity demonstrate that our benchmark extends beyond SceneFun3D and better aligns with real-world embodied task demands.

---

> > ### Comment · Reviewer_qb6v · 2025-08-04
> >
> > Thanks the author for the clarification. All of my concerns has been addressed. I appreciate the clarification between the task setting of the paper and SceneFun3D, which elevates the merit of this work. I'd appreciate it if the author can make the distriguishment clearer in the final version of the paper. I'll increase my score by 1.

---

> > > ### Author Response · Authors · 2025-08-05
> > > **Thank you for your response!**
> > >
> > > Thank you very much for your positive feedback and for increasing your score. We're glad to hear that your concerns have been fully addressed, and we truly appreciate your recognition of the distinction between our task setting and SceneFun3D. We will make sure to further clarify this distinction in the final version of the paper.

---

### Official Review · Reviewer_DXFg · 2025-07-03

**Clarity:** 2
**Significance:** 2
**Originality:** 3
**Rating:** 4
**Confidence:** 4

**Summary:**

This paper proposes a new task termed fine-grained 3D embodied reasoning, which jointly predicts 3D affordance masks, motion types, and motion axes in a 3D scene, conditioned on natural language instructions. To tackle this task, the authors introduce AffordBot, a framework that leverages surround-view synthesis and 3D-to-2D projection to transform point cloud input into a holistic multi-modal representation. A tailored chain-of-thought (CoT) paradigm then enables MLLM (e.g., Qwen2.5-VL-72B) to actively select a view and reason step-by-step over the scene. The approach achieves state-of-the-art performance on the SceneFun3D benchmark, demonstrating strong affordance grounding and motion reasoning capabilities.

**Questions:**

- How is the information from the geo-sem descriptor injected into the selected 2D views? Directly mark the segments and semantics on 2D images? Or represent in textual formats such as json?
- Why is AffordBot marked “without 2D input” in Table 1? Though starting with 3D point clouds, AffordBot takes as input 2D views and perceives the scene with a 2D visual encoder. Hence, I think AffordBot utilizes 2D input.
- Typo: Line 307 “agentS”

**Ethical Concerns:**

["NO or VERY MINOR ethics concerns only"]

**Final Justification:**

Technically solid paper. I lean towards accept.

**Limitations:**

Yes

**Quality:**

3

**Strengths And Weaknesses:**

#### Strength
- This paper proposes a new task that unifies affordance understanding and interaction estimation. The motivation is sound and the authors formulate this task well, representing the output with three joint parts: affordance grounding, motion type and motion axis.
- Well-designed multi-modal representation. The surround-view synthesis and adaptive 3D-to-2D projection pipeline successfully transforms 3D point cloud data into an MLLM-compatible format, avoiding the limitations of traditional video-based input and enabling richer spatial context.
- Extensive experiments and analyses. The authors conduct in-depth analyses such as on how different components contribute to the final performance, which shows the significance of enriched visual synthesis.

#### Weaknesses
- The motion axis is represented by a few text templates specifying the motion direction, which limits the expressiveness. And the prediction regarding such coarse templates can likely be handled simply by LLM’s capability of commonsense reasoning.
- The framework of AffordBot could confront cumulative error from the upstream segmentation module, i.e., the whole performance can be bottlenecked by the geo-sem descriptor. As shown in Table 4, the segmentation accuracy is quite low on some categories, which further consolidates the concern.
- For the proposed fine-grained 3D embodied reasoning task, I think the relationship to “embodied” is somewhat weak. It appears that the task does not require embodied awareness and can be handled by a non-embodied generic model. And for the proposed framework AffordBot, the only embodied-related part is enriched visual synthesis that generates views based on a specific anchor.

---

> ### Author Rebuttal · Authors · 2025-07-31
>
> Thank you for your valuable comments. We are pleased that the reviewer recognizes our contribution, particularly its clear motivation, well-designed method, and thorough experimental analysis. Please see our responses below.
>
> **W1: Expressiveness of the motion-axis templates.**
>
> The discrete motion-axis templates adopted in our framework serve as a directly executable semantic interface that covers common situations in the real physical world. Specifically, the majority of affordance elements (e.g., "drawer", "plug") are aligned with primary axes. Thus, discretizing motion directions into this set of semantic categories is not merely a simplification, but a design choice grounded in the spatial regularity of real-world environments. Moreover, the predicted motion in our framework functions as a high-level semantic signal—analogous to instructions like "pull outward horizontally". This abstraction enables our high-level system to focus on semantic reasoning, while the execution details will be handled by the downstream low-level action system. As reviewer #m6AC noted, this design provides a practical and interpretable bridge between multimodal LLMs and robotic control, and has also been validated by prior work such as [1]. We appreciate the reviewer’s insight and plan to extend the current design toward finer-grained primitives or continuous vectors in future work.
>
> **W2: Influence of segmentation quality.**
>
> Thank you for the insightful comment. We agree that upstream segmentation quality affects overall performance. To assess this, we conducted an oracle study using ground-truth masks, which improved AP50 from 20.0 to 39.4, highlighting potential gains from better segmentation. Despite current imperfections, our framework still significantly outperforms the previous SOTA Fun3DU. We plan to explore MLLM-based refinement to further enhance segmentation and will include this analysis in the revised version.
>
> | Upstream Method   | mAP  | AP50 | AP25 |
> |-------------------|------|------|------|
> | Mask3D proposals  | 15.5 | 20.0 | 23.3 |
> | GT proposals      | 35.7 | 39.4 | 45.4 |
>
> **W3: Embodied relevance.**
>
> We understand the reviewer's concerns about the degree of "embodiment". We would like to clarify that, by definition, our task inherently requires direct interaction with the real environment, which we believe constitutes a meaningful form of embodiment. Whether in terms of task conditions or application scenarios, the model must perceive and adjust decisions in real time within the physical world, represented by the 3D point cloud of the scene. This key distinction sets our approach apart from typical non-embodied visual question answering. Our task serves as the high-level semantic reasoning component in the robotic execution chain, with outputs directly interfacing with low-level action controllers to form a complete task pipeline. Although some sub-steps seem to be replaceable by non-embodied methods (e.g., OpenMask3D [2], Fun3DU [3]), as shown in Table 1 of the paper, their performance is significantly lower than ours. Our framework exhibits embodiment in several concrete ways. First, the surround-view synthesis and active view selection simulate the perception process of an embodied agent observing a real scene. Second, the Chain-of-Thought reasoning and prompt design operate on multi-modal inputs grounded in 3D space, enabling high-level semantic decisions based on spatial cues. Third, the model outputs target locations, motion types, and axes as interpretable signals that guide downstream action execution. Together, these components form an observation–reasoning–action chain tailored for embodied tasks, rather than passive visual understanding.
>
> **Q1: Explanations of geo-sem descriptor injection.**
>
> As illustrated in Section 3.2 (Main Paper) and Figure 4 (Supplementary), we first project each 3D element’s bounding box onto the rendered 2D view, producing a corresponding 2D region bounding box. Each 2D box is assigned a unique ID, ensuring a one-to-one correspondence between 3D elements and their 2D projections. We then convert the geometric (e.g., center coordinates, box size) and semantic (e.g., affordance type) information of each element into natural language and append it to the MLLM prompt. Instead of using a separate JSON file, we integrate all geo-sem descriptors as inline text alongside the image, which enables the language model to jointly reason over visual, geometric, and semantic information in a unified manner.
>
> **Q2: Clarifications of input modality.**
>
> We would like to clarify that our intent of "without 2D input" refers specifically to not relying on any **raw 2D input** provided by the dataset. AffordBot operates solely on 3D point cloud and generates all 2D views internally via online surround-view rendering. These rendered images serve as intermediate representations, not external inputs. To avoid confusion, we will revise the column header in Table 1 to read "2D raw input" for clarity.
>
> **Q3: Minor issues.**
>
> Thanks for your feedback, and we will correct this in the revised version.
>
> **References:**
>
> [1] Zawalski M, Chen W, Pertsch K, et al. Robotic control via embodied chain-of-thought reasoning[J]. arXiv preprint arXiv:2407.08693, 2024.
>
> [2] Takmaz A, Fedele E, Sumner R W, et al. Openmask3d: Open-vocabulary 3d instance segmentation[J]. arXiv preprint arXiv:2306.13631, 2023.
>
> [3] Corsetti J, Giuliari F, Fasoli A, et al. Functionality understanding and segmentation in 3D scenes[C]//Proceedings of the Computer Vision and Pattern Recognition Conference. 2025: 24550-24559.

---

> > ### Comment · Reviewer_DXFg · 2025-08-06
> >
> > Thanks for the authors' response. My concerns are addressed.

---

> > > ### Author Response · Authors · 2025-08-06
> > > **Thank you for your response!**
> > >
> > > Thank you for your response! We're glad to hear that all of your concerns have been resolved.

---

### Comment · Area_Chair_kpj9 · 2025-08-02
**Friendly Reminder: Engaging with Author Rebuttals**

Dear Reviewer,

Thank you for your time and expertise in reviewing for NeurIPS 2025. As we enter the rebuttal phase, we kindly encourage you to:

1) Read and respond to authors' rebuttals at your earliest convenience.

2) Engage constructively with authors by addressing their clarifications in the discussion thread.

3) Update your review with a "Final Justification" reflecting your considered stance post-rebuttal.

Your active participation ensures a fair and collaborative evaluation process. Please don’t hesitate to reach out if you have any questions.

With gratitude,

Your AC

---

### Decision · Program_Chairs · 2025-09-17

**Decision:**

Accept (poster)

**Comment:**

This paper introduces a new task, Fine-grained 3D Embodied Reasoning, which requires predicting a structured triplet (spatial location, motion type, motion axis) for affordance elements in a 3D scene based on a task instruction. Despite the noted limitations, this paper makes a clear contribution by defining a new and important task and providing a strong, well-validated baseline solution. The concerns about segmentation dependency and 2D projection are important for the discussion and future work, but do not diminish the paper's novelty, technical soundness, or significance for the NeurIPS community. All four reviewers hold positive ratings (Accept, Accept, Weak Accept, Weak Accept).